# D-ROMs and PAT Tests Reveal a High Level of Oxidative Stress in Patients with Severe Well-Controlled Asthma, and D-ROMs Are Positively Correlated with R20 Values That Indicate Approximate Central Airway Resistance

**DOI:** 10.3390/jpm13060943

**Published:** 2023-06-02

**Authors:** Ourania S. Kotsiou, Konstantinos Tourlakopoulos, Lamprini Kontopoulou, Georgios Mavrovounis, Ioannis Pantazopoulos, Paraskevi Kirgou, Sotirios G. Zarogiannis, Zoe Daniil, Konstantinos I. Gourgoulianis

**Affiliations:** 1Department of Respiratory Medicine, Faculty of Medicine, University of Thessaly, 41100 Biopolis, Greece; 2Faculty of Nursing, University of Thessaly, 45550 Gaiopolis, Greece; 3Department of Physiology, University of Thessaly, 41100 Biopolis, Greece

**Keywords:** asthma, impulse oscillometry, oxidative stress, spirometry

## Abstract

Background: The derivatives-reactive oxygen metabolites (d-ROMs) and plasma antioxidant capacity (PAT) tests are oxidative indexes. Severe asthma has been related to oxidative stress. We aimed to investigate d-ROMs and PAT values in severely controlled asthmatics and the correlation of these values with lung function. Methods: Blood samples were collected from severely controlled asthmatics and centrifuged at 3000 rpm for 10 min. The supernatant was collected. The assays were performed within three hours of collection. The fraction of exhaled nitric oxide (FeNO), impulse oscillometry (IOS), and spirometry were determined. Symptom control was recorded using the asthma control test (ACT). Results: Approximately 40 patients with severe controlled asthma (75%: women), mean age of 62 ± 12 years, were recruited. Approximately 5% had obstructive spirometry. The IOS revealed airway abnormalities even though the spirometric results were within the normal range, with it being more sensitive than spirometry. The D-ROMs and PAT test values were higher than normal, indicating oxidative stress in severe asthmatics with controlled asthma. D-ROMs were positively correlated with R20 values, indicating central airway resistance. Conclusions: The IOS technique revealed an otherwise hidden airway obstruction with spirometry. The D-ROMs and PAT tests revealed a high level of oxidative stress in severe controlled asthmatics. D-ROMs correlate with R20, indicating central airway resistance.

## 1. Introduction

The role of the imbalance between oxidants and antioxidants in favor of oxidants, termed oxidative stress in the pathogenesis of asthma, arises as the chicken or the egg causality dilemma. Although airway inflammation in asthma tends to increase the production of reactive oxygen species (ROS) and reactive nitrogen species (RNS), contributed to mainly by eosinophils and neutrophils [1], accumulating evidence suggests the involvement of ROS and RNS and oxidative stress in the genesis and modulation of asthma [2,3,4,5].

In the pathogenesis of asthma, eosinophilic and neutrophilic airway inflammation induces symptoms and oxidative stress [6]. Since eosinophils contain nicotinamide adenine dinucleotide phosphate (NADPH) oxidase, eosinophilic peroxidase (ΕPO), and nitric oxide (NO) synthesis, ROS can be generated in the asthmatic airway [6]. Moreover, ciliary dysfunction is increased in neutrophilic asthma associated with increased NADPH Oxidase 4 (NOX4) expression and is attenuated by NADPH oxidase inhibition.

The oxidative stress generated in the airway of patients with asthma causes epithelial cell injury [7]. Furthermore, ROS induce transforming growth factor-β1 (TGF-β1) secretion from epithelial cells, which may induce airway remodeling in asthma associated with severe forms of the disease [6,7,8,9].

The derivatives of reactive oxidative metabolites (d-ROMs) test analyzing the total amount of hydroperoxides in serum via the Fenton’s reaction [8] is used as an oxidative stress index, and the values correlate with the prognosis and mortality from cardiovascular events [10,11]. With the innovative plasma antioxidant capacity (PAT) test, we determine the total antioxidant capacity of the plasma through chromatography [12]. Both tests have been scarcely evaluated in severe asthma patients [2,9]. The normal values of the d-ROMs and PAT tests are: 250–320 U. Carr and 2200–2800 U. Carr, respectively [10,11,12,13]. 

Severe asthma has been associated with the mechanisms of oxidative biology. The utility of these tests in assessing the overall oxidative status of patients with asthma and the correlation of the oxidative stress values with lung function needs to be better evaluated. This study aimed to investigate the d-ROMs and PAT test levels in severely controlled asthmatics and the correlation of the oxidative stress values with lung function.

## 2. Materials and Methods

In this pilot study, blood samples were collected from patients with severely controlled asthma monitored at the Asthma Outpatient Clinic of the University of Thessaly from 1 November 2022 to 1 December 2022. 

### 2.1. Study Population 

Our sample consisted of patients between 18 and 80 years of age with severe well-controlled asthma requiring high-dose ICS/LABA combination therapy to prevent it from becoming uncontrolled. Asthmatic patients currently smoking or with a smoking pack history greater than 10 were excluded. We also excluded patients receiving systemic corticosteroids or suffering from COPD (i.e., chronic bronchitis or emphysema) and/or other relevant lung diseases causing alternating impairment in lung function or respiratory infection during the last month.

### 2.2. Ethics

Verbal and written informed consent were obtained from all subjects before the study, according to the Helsinki Declaration. The protocol was approved by the Larissa University Hospital Ethics Committee (approval number: 48841/25/10/2019).

### 2.3. Measurements

Symptom control was determined using the asthma control test (ACT). Moreover, detailed lung function tests were performed in the following order: the fraction of exhaled nitric oxide (FeNO), impulse oscillometry (IOS) performance, and then spirometry.

### 2.4. Asthma Control Test (ACT)

The ACT provided us with information on how well asthma had been controlled over the last four weeks, giving a score out of 25. A score of less than 20 means that asthma has not been controlled during the past four weeks [13].

### 2.5. Lung Function Tests

#### 2.5.1. Fraction of Exhaled Nitric Oxide (FeNO) 

The fraction of exhaled nitric oxide (FeNO) (MEDISOFT, MEDICAL GRAPHICS Corp., Saint Paul, MN, USA) was performed according to American Thoracic Society/European Respiratory Society (ATS/ERS) guidelines [14]. The cut-off values for normal FeNO levels were below 25 parts per billion (ppb) and elevated FeNO above 50 ppb [15,16].

#### 2.5.2. Impulse Oscillometry (IOS)

IOS measurements were performed in triplicate according to standard guidelines with a Jaeger MasterScreen IOS system (Carefusion, Wurmlingen, Germany, JLAB software version 5.22.1.50) [17,18,19]. Volume calibration was performed daily using a 3-L volume syringe, and the accuracy of resistance measurements was confirmed daily using a standard 0.2 Kpa.s.L−1 resistance mesh. Each patient was seated upright, wore a nose clip, and pressed on their cheeks with their hands to prevent an upper airway shunt [20,21,22]. Mean values for resistance at 5 Hz (R5: indicates total airway resistance); resistance at 20 Hz (R20: approximates central airway resistance); reactance at 5 Hz (X5: relates to compliance); resonant frequency (Fres); and integrated area of low-frequency X (AX) values were derived as previously reported [22,23,24,25]. Fres and AX are values that detect expiratory flow limitations. R5–R20 and AX are thought to reflect changes in the quiet zone of the lungs [26]. Cut-off pathological values for R5 higher than 0.5 kPa/L/s and AX higher than 1.0 kPa/L were suggested for use [26]. The normal value of Fres in adults is 7–12 Hz [26].

#### 2.5.3. Spirometry

Spirometry (Spirolab, MIR, Rome, Italy) was performed according to ATS/ERS guidelines [27]. The predicted values were derived from the Global Lung Initiative [28]. To avoid any negative effects of forced expiration on the airway, spirometry was never performed before the IOS measurements. The percent predicted forced vital capacity (%FVC), the percent predicted forced expiratory volume in 1 s (%FEV1), and the FEV1/FVC ratio were obtained. The best of at least three technically acceptable results was selected.

#### 2.5.4. Blood Sampling

Blood samples were collected. The samples were centrifuged at 3000 rpm for 10 min at 4 °C, and the supernatant was collected. Assays were performed within three hours of blood collection.

#### 2.5.5. Measurement of D-ROMs

d-ROMs FAST tests were performed on fresh serum within three hours of blood collection. No previously frozen and then defrosted samples were used for the test. In this test, hydroperoxides released from proteins in the acidic medium are converted to alkoxy and peroxyl radicals in the presence of transition metals, which are able to oxidize alkylsubstituted aromatic amines (N,N-diethylparaphenylendiamine), showing increased absorbance at 505 nm. The method detects the density of the colored complex photometrically, which is directly proportional to the concentration of hydroperoxides, as previously described [29,30]. D-ROMs values from 401 up to 500 U.Carr and above 500 U.Carr indicate a high and a very high level of oxidative stress, respectively [29,30].

#### 2.5.6. Measurements of the PAT Test

The PAT test (OB manager, FRAS5, H&D, Parma, Italy) is an automated test designed to assess the antioxidant capacity of plasma by measuring its ferric-reducing ability. Ferric to ferrous ion reduction causes a color change that can be photometrically assessed, as previously described [12]. PAT test values above 2800 μmol/L indicate a very high level of oxidative stress [12].

#### 2.5.7. Statistical Analyses

The chi-square test was used to make comparisons between frequencies. An unpaired t-test was used for comparing parametric data between two groups, while non-parametric data were analyzed with the Mann–Whitney U test. Spearman’s correlation was used for correlation analysis between oxidative markers, pulmonary function variables, and demographic, clinical, and laboratory parameters. Statistical analyses were performed with IBM SPSS Statistics for Windows, version 23.0, IBM Corp., Armonk, NY, USA. 

## 3. Results

Forty asthmatics with severely controlled asthma were recruited for the study with an average age of 62 ± 12 years, 55% of whom were women. The mean asthma history, maintenance therapy, lung function parameters, d-ROMs and PAT tests values, and ACT score of the study population and according to gender are presented in Table 1. 

All patients received high doses of ICS/LABA and LAMA. Approximately 5% of the study population had obstructive spirometry. The mean R5 values were higher than 0.5 kPa/L/s in the studied population. The mean AX values were abnormal as they were higher than 1.0 kPa/L; Fres levels were also elevated in the population. The males had significantly higher R5 values which indicated higher total airway resistance, higher R20 values which indicated approximate central airway resistance, and Fres and AX values which revealed higher expiratory flow limitations than females. The males were significantly taller (1.75 m vs. 1.50 m, *p* = 0.030) than the females. There was no significant difference in weight between the genders (85 vs. 75 kg, *p* = 0.125). Interestingly, the IOS results revealed airway abnormalities even though the spirometry results were within range, thus being more sensitive than spirometry. A significant negative correlation was found between FEV1 and R5 (r = −0.547, *p* = 0.001), di5-20 (r = −0.645, *p* = 0.001), and Fres (r = −0.613, *p* < 0.001).

The D-ROMs and PAT test values indicated very high levels of oxidative stress given the reference values (Figure 1). We found that d-ROMs were positively correlated with R20 values that indicate central airway resistance (r = 0.423, *p* = 0.016). No correlation was found between d-ROMs, FeNO, the ACT test, asthma history, and maintenance therapy. No correlation was found between the PAT test values and the several lung function parameters, neither between FeNO nor the ACT test. No correlation was found between the PAT test values, asthma history, and maintenance therapy. 

## 4. Discussion

In this study, we investigated d-ROMs and PAT test values in severely controlled asthmatics and the correlation between oxidative stress values and lung function. The D-ROMs and PAT test values indicated a very high level of oxidative stress in severe asthmatics with controlled asthma. Interestingly, d-ROMs were positively correlated with R20 values that indicate central airway resistance. No correlation was found between the tests and the spirometric parameters. Moreover, we found that the IOS results revealed airway abnormalities even though the spirometric results were within range, thus being more sensitive than spirometry. Moreover, a significant inverse correlation was found between FEV1 and R5, di5-20, and Fres.

ROS are constitutively present in cells and tissues in small but measurable concentrations [11,31]. They act as guidance cues during embryonic development [11,31] and as signaling molecules activating relevant biological and physiological pathways [10,32]. However, an increase in ROS production, such as during an altered metabolic state or as a consequence of chronic exposure to external/environmental factors, may have harmful effects [11]. The oxidant “burst” in asthma is probably a nonspecific process due to the concomitant action of several inflammatory pathways. The endogenous sources of ROS are the endoplasmic reticulum of immune cells such as phagocytes, activated eosinophils and neutrophils, monocytes, macrophages, and structural cells (epithelial cells, smooth muscle cells, and endothelial cells) or enzymes and enzymatic complexes (e.g., NO synthase, NADPH oxidases, and xanthine oxidase), mitochondria, and peroxisomes [33,34]. At the same time, ROS can be generated by exogenous sources, such as pollutants, allergens, ozone, cigarette smoke, organic solvents, metals, ultraviolet light, ionizing radiation, and some drugs [33,34]. Both endogenous and exogenous promoters or stimuli of ROS are involved in initiating and activating intracellular signaling, promoting inflammatory and immunological mechanisms in the airways. 

The most prominent ROS are the superoxide anion (O_2_), hydrogen peroxide (H_2_O_2_), and all peroxides (ROOR) [35]. The key enzymes that produce ROS/RNS in the lungs include NADPH oxidase, MPO, xanthine oxidase, and NO synthase (NOS) [33,34]. Both eosinophils and neutrophils have increased ROS-scavenging enzyme expression.

The amount of biologically active ROS might indicate individual health status, and their quantification holds potential for clinically relevant applications. Organic peroxides and lipoperoxides (d-ROMs) are intermediate metabolites generated early in the oxidative cascade [36], with them being a better indicator of early stages of oxidative-stress damage that might provide an opportunity for timely intervention [10]. 

Notably, several studies support that in cardiovascular patients, d-ROM values above 395–482 U.Carr are associated with an increase in all cardiovascular events, major adverse cardiovascular events, heart failure, atrial fibrillation, and cardiovascular death [11,37,38,39,40,41,42]. This study found that severe asthmatics had a remarkably higher mean value of d-ROMs at 531 ± 120 U.Carr than previous studies on patients with cardiovascular diseases, indicating a very high level of oxidative stress. It is required to investigate further the d-ROMs value cut-offs associated with higher morbidity and mortality risks in asthmatics and establish reference ranges. Although the relevance of d-ROMS to known cardiovascular risk factors is increasingly being recognized, there are scarce data regarding the d-ROMs value cut-offs in severe asthmatics. There is a previous report by Suzuki et al., who reported that acute exacerbations of asthma are associated with increased oxidative stress., as indicated by the serum ROM levels that would partly reflect eosinophilic and neutrophilic inflammation [2]. 

In this study, the mean PAT values in the group of asthmatics were 2807 ± 1344 U.Cor, indicating very high values of oxidative stress. The study of Petrushevska et al. found that the PAT test values were significantly higher in patients with severe COVID-19 than non-infected individuals (3048 ± 100.1 vs. 2406 ± 71.55 U.Cor, *p* < 0.001) [43]. 

For the first time, our study showed that d-ROMs were positively correlated with R20hz values, an indicator of approximate central airway resistance, demonstrating that oxidative stress is associated with altered pulmonary mechanics. A previous study by Sunil et al. found that World Trade Center dust exposure, at 21 days, increased lung resistance, central airway resistance, tissue damping, and tissue elastance along with inflammation and oxidative stress in a murine model [44]. No correlation was found between PAT test values and lung function parameters. The relatively small sample size may not have the statistical power to expose a significant effect.

Furthermore, no correlation was detected between d-ROMs or PAT values and FeNO. Similarly, the study by Sone et al. found that there was no significant association between the levels of salivary NO and oxidative stress markers such as d-ROMs and biological antioxidant potential (BAP) in a sample of 250 elite university athletes in Japan [45]. The FeNO levels were low enough among the studied subjects; thus, the relationship between the levels of oxidative stress markers and FeNO may be difficult to confirm owing to the small individual differences in this cross-sectional study.

For many years, IOS was promoted as an alternative measurement technique for airway resistance and obstruction in patients where maneuvers involved in plethysmography and spirometry prove difficult to perform, such as in the elderly, children, and obese patients [46,47,48]. IOS parameters are obtained more easily and effortlessly in comparison to FEV1. There are also data supporting that IOS resistance parameters have good long-term repeatability in asthmatics [49]. In this study, we highlight the superiority of IOS to reveal airway abnormalities in controlled severe asthmatics compared to spirometry, as the mean R5, AX, and Fres levels were elevated in the population of this study. Accordingly, there are several studies supporting higher sensitivity with IOS than spirometry for peripheral airway restriction [50]. Cottini et al. supported that IOS-defined small airway disease which is overwhelmingly present across asthma severities may be overlooked by standard spirometry, especially in milder asthma [51]. Moreover, a higher sensitivity, negative predictive value, and earlier response of methacholine with IOS have been documented, compared to methacholine with spirometry [52,53]. 

We also found an inverse correlation between IOS parameters and FEV1. In support of our findings, there are many previous studies that have supported a moderate inverse correlation between IOS parameters such as R5Hz, Fres, and AX and FEV1 variables [54,55,56,57]. More specifically, Nair et al. demonstrated that 1 unit change in %FEV1 corresponds to a 2.5% change in %R5 [57].

In this study, we found a significant difference in the impedance values between the genders. However, the differences were in the AX and Fres results which were not corrected for anthropometric parameters, and further studies are needed before safe conclusions can be drawn. According to Robinson et al., asthma is associated with abnormalities in IOS measures of peripheral airway dysfunction, which is stronger in men and in those with asthma persisting since childhood [58]. Porojan-Suppini N et al. also supported that prediction equations were influenced by the height of the subjects and also by gender [59]. Moreover, with increasing BMI, resistance and reactance increased compatible with lung and airway compression from mass loading [60]. In this study, the men were significantly taller (1.75 m vs. 1.50 m, *p* = 0.030) than the females, thus potentially confounding the detected difference in the impedance values between the genders.

This study has some limitations. Firstly, the small sample size and the high mean age of the subjects reduced the power of the study and the degree to which we can apply the results of this study to a broader context. Secondly, a major limitation was that the sample size consisted only of patients with severe controlled asthma and lacked a control group of healthy individuals. Moreover, only 5% of the study population had obstructive spirometry. Moreover, in this study, we did not report Z-scores, which are believed to be more reliable indicators for interpreting the oscillometry results [61]. However, S. Salvi and co-workers argued that Z-scores are not appropriate to be used in interpreting respiratory oscillometry results given that oscillometric indices are not normally distributed; thus, Z-score may not accurately reflect the probability of being within the healthy distribution, and secondly, the anthropometric factors (such as height and gender) have a small impact on impulse oscillometry indices (the highest determination coefficient of the predictive equation is ∼0.20), and they supported that using fixed cut-offs for diagnosis may be a simple and feasible alternative [62]. Furthermore, the correlation between R20 and d-ROMs might be influenced by height, weight, or sex. Some studies have reported a potential association of d-ROMS with gender (higher values than females) and obesity, probably due to lifestyle-related parameters reflecting mainly an unbalanced diet. However, in our population, there were no statistical differences in the female: male ratio and weight differences between gender; there was only a height difference between the genders [63,64]. Finally, the correlation of d-ROMs was with R20 and not R20%, which is corrected for anthropometric parameters, and therefore, it is not a clear indication of alteration of the airways. However, this pilot study is one of the essential stages in a research project before a large-scale study, which allows us to refine the study design, thus increasing the validity. It would be very interesting to investigate oxidative levels and correlations with lung function parameters in patients with different stages of asthma severity and uncontrolled disease in a future study by including a larger group of patients with airway obstruction.

## 5. Conclusions

The IOS technique is increasingly employed in routine lung function testing, revealing an otherwise hidden airway obstruction with spirometry. A significant inverse correlation was confirmed between FEV1 and IOS parameters of airway obstruction such as R5, di5-20, and Fres. For the first time, the D-ROMs and PAT tests indicated a high level of oxidative stress in severe asthmatics with controlled asthma. The D-ROMs values were correlated with central airway resistance. Future studies are required to validate the scope and potential significance of oxidative status measurements and their correlations with airway mechanics in larger cohorts to establish multi-ethnic reference ranges and standardize the commercially available oxidative status tests prior to their routine incorporation into clinical practice.

## Figures and Tables

**Figure 1 jpm-13-00943-f001:**
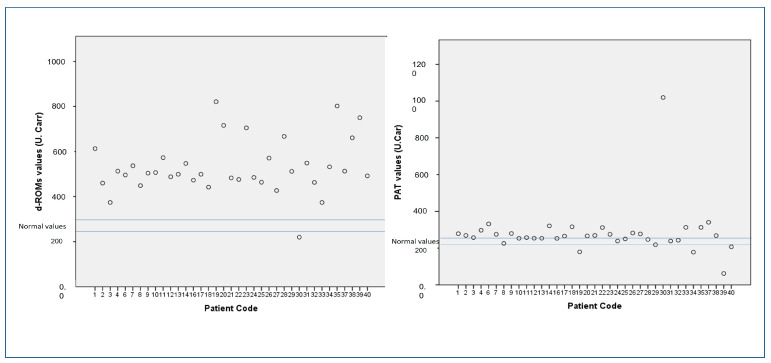
d-ROMs and PAT values in the study population.

**Table 1 jpm-13-00943-t001:** The main characteristics of the study population (N = 40) and according to gender.

Parameters	All (n = 40)	Males (n = 18)	Females (n = 22)	*p*-Value
Age (years)	62 ± 12	67 ± 9	61 ± 13	0.244
Mean asthma history (years)	14 ± 10	10 ± 8	18 ± 3	0.320
Maintenance therapy				
ICS/LABA, n (%)	40 (100)	18 (100)	22 (100)	0.720
Montelukast, n (%)	5 (13)	2 (11)	3 (14)	0.360
LAMA, n (%)	40 (100)	18 (100)	22 (100)	0.740
Anti-IL-5, n (%)	7 (18)	4 (22)	3 (14)	0.200
Anti-IgE, n (%)	2 (5)	1 (6)	1 (5)	0.400
FeV1/FVC	74 ± 11	71 ± 13	74 ± 10	0.548
FeV1 (%pred)	93 ± 19	84 ± 19	97 ± 18	0.299
FVC (%)	109 ± 18	110 ± 28	109 ± 15	0.939
R5 (Kpa.s.L^−1^)	3 ± 14	14.5 ± 7	0.4 ± 0.2	**0.011**
R5Hz (%)	121 ± 43	136 ± 37	117 ± 45	0.304
X5 (Kpa.s.L^−1^)	−0.1 ± −0.06	−0.11 ± 0.3	−0.11 ± 0.1	0.633
X5 (%)	−200 ± 10	−709 ± 28	−145 ± 45	0.135
Fres (Hz)	147 ± 35	552 ± 15	18 ± 8	**0.019**
AX(Hz kPa s L^−1^)	12 ± 6	46 ± 21	1 ± 1	**0.045**
R20 (Kpa.s.L^−1^)	3 ± 1	9 ± 7	0.3 ± 0.1	**0.009**
R20Hz (%)	112 ± 31	120 ± 37	110 ± 29	0.454
d-ROMs (U.Carr)	531 ± 120	486 ± 163	548 ± 98	0.167
PAT (U.Cor)	2807 ± 1344	3483 ± 2378	2557 ± 553	0.062
ACT	22 ± 1	23 ± 2	22 ± 1	0.367
FeNO (ppb)	10 ± 6	11 ± 8	10 ± 6	0.786

Abbreviations: ACT, asthma control test; AX, area of low-frequency X; d-ROMs, derivatives-reactive oxygen metabolites; FeV1, forced expiratory volume in 1 s; fres, resonant frequency; FVC, force vital capacity; ICS, inhaled corticosteroids; Ig, immunoglobin; IL, interleukin; LABA, long-acting bronchodilators; LAMA, long-acting muscarinic antagonists; PAT, plasma antioxidant capacity; R5, resistance at 5 Hz, R20, resistance at 20 Hz; X5, reactance at 5 Hz.

## Data Availability

The data that support the findings of this study are available from the corresponding author, O.S.K., upon request.

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
