# Peer review of "D-ROMs and PAT Tests Reveal a High Level of Oxidative Stress in Patients with Severe Well-Controlled Asthma, and D-ROMs Are Positively Correlated with R20 Values That Indicate Approximate Central Airway Resistance"

_jpm, 2023, doi:10.3390/jpm13060943_

Round 1

Reviewer 1 Report

1) The title seems to be misleading with respect to the goal and the results.
2) The introduction does not seem to be sufficient to support the aim and results of the study.
In fact, in the introduction there is much argument of the D-ROM and PAT tests, as well as in
the title, but the results do not seem to be significant from this point of view apart from the
correlation with IOS.
3) Reference 11 does not appear to be in line with 52-53.
4) Reference 11 is the same as reference 31.
5) Methods lack a control group.
6) In the discussion the researchers comment that the values of D-ROM and PAT are high in
patients with asthma, as also indicated in the title, but the results do not seem significant. In
general, the discussion seems too misleading and confusing with the objective and results of
the study, and seems not to be well-connected with the study hypothesis, especially in lines
189-243. Could this statement be better restated in the debate?
7) In line 244 researchers state that for the first time they show a correlation with R20hz, is
this enough to say that there is a correlation with lung function? Moreover, there is not much
describes in the discussion of another objective, that is that in asthmatics the oxidative stress
values have been high in the D-ROM and PAT tests. It might also be useful to insert a figure.
8) In the discussion you could comment more on the result that D-ROM, as indicated in
reference 57 (lines 249-252), does not correlate with FeNo, and how can this be explained,
given that oxidative stress, calculated by D-ROM and PAT, is high in patients with asthma?
9) In the conclusion missing "time" after “for the first” in line 270.

Author Response

Response to Reviewer 1 Comments:

  • The title seems to be misleading with respect to the goal and the results.

Response: Thank you for this comment. The normal values for d-ROMs and PAT tests are 250-320 U.Carr and 2200-2800 U.Cor, respectively. In this study, we found that the mean d-ROMs and PAT levels of the study population were higher than the normal values. More specifically, the values were 531±120 U.Carr and 2807±1344 U.Cor, respectively, supporting that these tests reveal a high level of oxidative stress in patients with severe well-controlled asthma. Given that only D-ROMs values were correlated with central airway resistance the title has now been revised, as suggested.

  • The introduction does not seem to be sufficient to support the aim and results of the study. In fact, in the introduction there is much argument of the D-ROM and PAT tests, as well as in the title, but the results do not seem to be significant from this point of view apart from the correlation with IOS.

Response: Thank you for this comment. In the introduction, we discussed the rationale behind the aim of this study. Information has been revised regarding previous results on that issue. We hope these revisions meet your expectations.

  • Reference 11 does not appear to be in line with 52-53.

Response: Thank you for this point. We have corrected this error made by mistake.

  • Reference 11 is the same as reference 31.

Response: Thank you for this point. We have corrected this error made by mistake.

  • Methods lack a control group.

Response: Thank you for this comment. We referred as a major limitation of this study that the sample size consisted only of patients with severe controlled asthma and lacked a control group of healthy individuals. However, this pilot study is one of the essential stages in a research project before a large-scale study, which allows to refine the study design thus increasing the validity.

  • In the discussion the researchers comment that the values of D-ROM and PAT are high in patients with asthma, as also indicated in the title, but the results do not seem significant.

Response: Thank you for this comment. Αs we previously mentioned, the normal values for d-ROMs and PAT tests are 250-320 U.Cor and 2200-2800 U.Carr, respectively. In this study, we found that the mean d-ROMs and PAT levels of the study population were 531±120 U.Carr and 2807±1344 U.Cor, respectively, higher than the normal values, supporting that these tests reveal a high level of oxidative stress in patients with severe well-controlled asthma.

  • In general, the discussion seems too misleading and confusing with the objective and results of the study, and seems not to be well-connected with the study hypothesis, especially in lines 189-243. Could this statement be better restated in the debate?

Response: Thank you for this comment. In the revision, we discussed the results of the study by following a logical order according to the presented objectives.

  • In line 244 researchers state that for the first time they show a correlation with R20hz, is this enough to say that there is a correlation with lung function?

Response: Thank you for this remark. We found that d-ROMs were positively correlated with R20 values which is a sensitive marker that indicate central airway resistance, i.e. a lung function abnormality.

  • Moreover, there is not much describes in the discussion of another objective, that is that in asthmatics the oxidative stress values have been high in the D-ROM and PAT tests. It might also be useful to insert a figure.

Response: Thank you for this suggestion. In the revision, we have inserted a figure to better describe the high oxidative stress values in the D-ROM and PAT tests.

  • In the discussion you could comment more on the result that D-ROM, as indicated in
    reference 57 (lines 249-252), does not correlate with FeNo, and how can this be explained, given that oxidative stress, calculated by D-ROM and PAT, is high in patients with asthma?

Response: Thank you for the comment. However, the FeNO levels were low enough among the studied subjects; thus, the relationship between the levels of oxidative stress markers and FeNO may be difficult to confirm owing to the small individual differences in this cross-sectional study.

  • In the conclusion missing "time" after “for the first” in line 270.

Response: Thank you for this point. In the revised manuscript, we corrected this error.

We appreciate you taking the time to offer us your insights related to the paper. We found your feedback very constructive. We tried to be responsive to your concerns.

Reviewer 2 Report

In this study, in a small sample of subjects affected by  bronchial asthma, clinically well controlled and without spirometry abnormalities, the authors found an impressive increase of some oxidative indexes in blood. This result may be interpreted as the demonstration of some biological activity of asthma  also when the most popular tests used in the clinical setting to monitor the disease are perfectly normal. Unfortunately, in this manuscript the authors do not give informations about the long term clinical history of the subjects, their maintenance therapy, if any and so on. We only know  that, when they have been studied, the subjects  were clinically stable with normal spirometry.  I am not surprised that some parameters measured with IOS were abnormal even with normal spirometry. In fact, some low sensitivity of FEV1 to pathologies affecting the peripheral airways is well known. Given the small number of subjects studied, I think the authors should supply more  informations about the clinical history of the subjects in order to allow a more accurate interpretations of the results. For example, the mean age of the subjects was quite high for a study about asthma. Given that situation, may we think that the results of this study only apply to subjects with a very long history of asthma or to old subjects affected by asthma whatever its duration might be? 

Author Response

Response to Reviewer 2 Comments:

  • In this study, in a small sample of subjects affected by bronchial asthma, clinically well controlled and without spirometry abnormalities, the authors found an impressive increase of some oxidative indexes in blood. This result may be interpreted as the demonstration of some biological activity of asthma also when the most popular tests used in the clinical setting to monitor the disease are perfectly normal.

Response: We sincerely thank you for your kind words about our paper. We are delighted to receive positive feedback from you.

  • Unfortunately, in this manuscript the authors do not give informations about the long term clinical history of the subjects, their maintenance therapy, if any and so on. We only know that, when they have been studied, the subjects were clinically stable with normal spirometry.  I am not surprised that some parameters measured with IOS were abnormal even with normal spirometry. In fact, some low sensitivity of FEV1 to pathologies affecting the peripheral airways is well known. Given the small number of subjects studied, I think the authors should supply more informations about the clinical history of the subjects in order to allow a more accurate interpretations of the results.

Response: Thank you for the comment. In the revised manuscript, we have given information about the long-term clinical history of the subjects and their maintenance therapy, as suggested.

  • For example, the mean age of the subjects was quite high for a study about asthma. Given that situation, may we think that the results of this study only apply to subjects with a very long history of asthma or to old subjects affected by asthma whatever its duration might be? 

Response: Thank you for the comment. We agree with you that the mean age of the subjects was quite high. The small sample size and the high mean age of the subjects reduced the power of the study and the degree to which we can apply the results of this study to a broader context. Hence, in the revision we report this fact as a study limitation.

We appreciate you taking the time to offer us your insights related to the paper. We found your feedback very constructive. We tried to be responsive to your concerns.

Reviewer 3 Report

Kotsiou et al. reported D-ROM and PAT test values and lung function parameters in severe well-controlled asthmatics. The topic and the data are interesting. However, there are incongruencies in the reported numbers, in data interpretation, and the discussion is not supported by the provided results.

Abstract: 75% females. Table 1: 22 females (therefore 55%)

Table1. “sum” could be misleading, “all” may be a more accurate word. What Tif is? fres and AX have the wrong units. The difference in some IOS parameters between males and females is impressive. Impedance values should be more dependent on height than sex. Are there differences in height or weight between males and females? Why are data reported according to gender? How were the % predicted IOS parameters computed?

Results: It is not clear why the results are divided by gender and how this supports the discussion. Only 2 out of the 5 spirometric parameters mentioned in the methods are reported. Also, the correlations between variables that are the most underlined in the introduction and title are quickly dismissed.

Some sentences should be moved to the discussion as they already report an interpretation of the data.

Line 169. wrong interpretation. R5 %predicted is not higher. Higher R5 values may be due to anthropometric parameter differences, not obstruction.

Discussion: Line 183: why? There is any data about this in the results? How many subjects have IOS parameters outside the range of normality?

The main result seems to be : d-ROMs were positively correlated with R20 values that indicate approximate central airway resistance.

However, the correlation is with R20 and not R20%. So it is not corrected for anthropometric parameters and therefore, it is not a clear indication of alteration of the airways. It is not the “quantity of R20 abnormality” that correlates with d-ROMs.

Author Response

Response to Reviewer 3 Comments:

  • Kotsiou et al. reported D-ROM and PAT test values and lung function parameters in severe well-controlled asthmatics. The topic and the data are interesting. However, there are incongruencies in the reported numbers, in data interpretation, and the discussion is not supported by the provided results.

Response: Thank you for the comments. We greatly appreciate that you find the topic and our data interesting. In the revision, we tried to be responsive to your concerns.

  • Abstract: 75% females. Table 1: 22 females (therefore 55%)

Response: Thank you for this point. In the revision we corrected this error made by mistake.

  • “sum” could be misleading, “all” may be a more accurate word.

Response: Thank you for this point. In the revision we have replaced the word, as suggested.

  • What Tif is?

Response: Thank you for this point. In the revision we have replaced the word, which is the FeV1/FVC ratio.

  • fres and AX have the wrong units.

Response: Thank you for this point. In the revision we have corrected the units.

  • The difference in some IOS parameters between males and females is impressive. Impedance values should be more dependent on height than sex. Are there differences in height or weight between males and females? Why are data reported according to gender?

Response: Thank you for this great comment. According to Robinson et al., asthma is associated with abnormalities in IOS measures of peripheral airway dysfunction, which is stronger in men and those with asthma persisting since childhood [Robinson PD, King GG, Sears MR, Hong CY, Hancox RJ. Determinants of peripheral airway function in adults with and without asthma. Respirology. 2017 Aug;22(6):1110-1117]. Porojan-Suppini N et al also supported that prediction equations were influenced by the height of the subjects and also by the gender (Porojan-Suppini N, Fira-Mladinescu O, Marc M, Tudorache E, Oancea C. Lung Function Assessment by Impulse Oscillometry in Adults. Ther Clin Risk Manag. 2020 Nov 26;16:1139-1150. doi: 10.2147/TCRM.S275920). Moreover, with increasing BMI, resistance and reactance increased compatible with lung and airway compression from mass loading (Berger KI, Wohlleber M, Goldring RM, Reibman J, Farfel MR, Friedman SM, Oppenheimer BW, Stellman SD, Cone JE, Shao Y. Respiratory impedance measured using impulse oscillometry in a healthy urban population. ERJ Open Res. 2021 Mar 29;7(1):00560-2020. doi: 10.1183/23120541.00560-2020). In this study, men were significantly higher (1.75m vs 1.50m, p=0.030). No significant difference was detected in weight between males and females (85 vs 75kg, p=0.125). In this study, men were significantly higher (1.75m vs 1.50m, p=0.030) than females potentially confounding the detected difference in impedance values found between women and men.

  • How were the % predicted IOS parameters computed?

Response: Thank you for the question. The % predicted IOS parameters were computed automatically by the device.

  • Results: It is not clear why the results are divided by gender and how this supports the discussion.

Response: Thank you for this comment. The results were divided by gender given that there were significant differences in IOS parameters, as previously discussed.

  • Only 2 out of the 5 spirometric parameters mentioned in the methods are reported.

Response: Thank you for the comment. We fixed this issue in the revised manuscript.

  • Also, the correlations between variables that are the most underlined in the introduction and title are quickly dismissed.

Response: Thank you for the comment. The title and discussion have now been revised to present more clearly the main findings of the study, as suggested.

  • Some sentences should be moved to the discussion as they already report an interpretation of the data.

Response: Thank you for the comment. The discussion has been revised accordingly.

  • Line 169. wrong interpretation. R5% predicted is not higher. Higher R5 values may be due to anthropometric parameter differences, not obstruction.

Response: Thank you for the comment. We correct this parameter.

  • Discussion: Line 183: why? There is any data about this in the results? How many subjects have IOS parameters outside the range of normality?

Response: Thank you for the comment. In the revision, we discuss this issue on lines 171-176, page 4. That addition improves the quality of our paper.

  • The main result seems to be: d-ROMs were positively correlated with R20 values that indicate approximate central airway resistance.

Response: Thank you for the remark. The title has been revised accordingly.

  • However, the correlation is with R20 and not R20%. So it is not corrected for anthropometric parameters and therefore, it is not a clear indication of alteration of the airways. It is not the “quantity of R20 abnormality” that correlates with d-ROMs.

Response: Thank you for this great comment. We discuss this issue as a limitation of this pilot study.

We appreciate you taking the time to offer us your insights related to the paper. We found your feedback very constructive. We hope the revisions meet your high standards.

Round 2

Reviewer 3 Report

The manuscript is improved. However, several concerns remain. There are questionable sentences and the significance/impact and the novel message of the manuscript are unclear.

For example:

- “Cut-off pathological values for R5 higher than 0.5 kPa/L/s, R5% higher than 150%, AX higher than 1.0 kPa/L were suggested for use [26]. The normal value of fres in adults is 7-12 Hz [26].”

Ref 26 seems inappropriate. R5% higher that 150% is not considered abnormal. Abnormality depends on the upper limit of normality and may correspond to different R5% according to anthropometrics as for technical standards for oscillometry ( King GG, et all. Technical standards for respiratory oscillometry. Eur Respir J. 2020 Feb 27;55(2):1900753. doi: 10.1183/13993003.00753-2019. )

- “Moreover, we found that IOS revealed airway abnormalities even though spirometric results were within range”. It is unclear if IOS values were abnormal or not. Zscores or the number of abnormal parameters are not reported.

- “In this study, we found a significant difference in impedance values between gender.” R5%, X5%, and R20% are the same. Differences are in AX and Fres that are not corrected for anthropometric parameters and Fres presents unreasonable values for males. This does not support the utility of comparing males and females.

- It is still not clear which is the meaning/the message of the R20 correlation with d-ROMs… it may just be that R20 and d-ROMs are similarly influenced by height, weight, or sex.

Author Response

Response to Reviewer 1 Comments:

  1. The manuscript is improved.

RESPONSE: Τhank you for the comment. We are glad to receive positive feedback from you.

  1. However, several concerns remain. There are questionable sentences and the significance/impact and the novel message of the manuscript are unclear. For example:- “Cut-off pathological values for R5 higher than 0.5 kPa/L/s, R5% higher than 150%, AX higher than 1.0 kPa/L were suggested for use [26]. The normal value of fres in adults is 7-12 Hz [26].” Ref 26 seems inappropriate.

RESPONSE: Thank you for the comment. Lipworth B et al. reported that abnormal values for peripheral resistance on impulse oscillometry (R5–R20 >0·15 kPa/L.s]) and reactance area (AX >0·93 kPa/L) were highly predictive of disease control (ref.26). Moreover, later, again, Lipworth B, et al., reported pragmatic abnormal IOS values as R5 > 0.5 kPa/l.s, R5–R20 > 0.1 kPa/l/s and AX > 1.0 kPa/l, in terms of being indicative of a clinically relevant degree of airflow obstruction. [Lipworth B et al. Normal spirometry equates to normal impulse oscillometry in healthy subjects. Respir Res. 2021 Mar 31;22(1):96.2]

  1. R5% higher that 150% is not considered abnormal. Abnormality depends on the upper limit of normality and may correspond to different R5% according to anthropometrics as for technical standards for oscillometry ( King GG, et all. Technical standards for respiratory oscillometry. Eur Respir J. 2020 Feb 27;55(2):1900753. doi: 10.1183/13993003.00753-2019. )

RESPONSE: Thank you for the comment. We corrected this issue.

  1. “Moreover, we found that IOS revealed airway abnormalities even though spirometric results were within range”. It is unclear if IOS values were abnormal or not. Zscores or the number of abnormal parameters are not reported.

RESPONSE: Thank you for the comment. We add this issue as a limitation of our study. However, S. Salvi and co-workers argued that Z-scores are not appropriate to be used in interpreting respiratory oscillometry results for the following reasons. First, oscillometric indices are not normally distributed; thus, the Z-score may not accurately reflect the probability of being within the healthy distribution. Secondly, although anthropometric factors (such as height and gender) have an impact on impulse oscillometry indices, their impact is not as large as that of spirometric indices (the highest determination coefficient of the predictive equation is 0.20). In this case, using fixed cut-offs for diagnosis may be a simple and feasible alternative. (Liang X, Zheng J, Wang Z, Gao Y. Reply to: Interpreting lung oscillometry results: Z-scores instead of fixed cut-off values? ERJ Open Res. 2023 Mar 27;9(2):00718-2022).

  1. “In this study, we found a significant difference in impedance values between gender.” R5%, X5%, and R20% are the same. Differences are in AX and Fres that are not corrected for anthropometric parameters and Fres presents unreasonable values for males. This does not support the utility of comparing males and females.

RESPONSE: Τhank you for this remark. We have corrected this issue in the manuscript found on page 7, lines 293-295.

- It is still not clear which is the meaning/the message of the R20 correlation with d-ROMs… it may just be that R20 and d-ROMs are similarly influenced by height, weight, or sex.

RESPONSE: There are studies reported a potential association of d-ROMS with gender (higher values over females) and in obesity, probably due to lifestyle-related parameters reflecting an unbalanced diet. However, in our population there were no statistical differences in the female:male ratio and weight differences between gender, there was only a height difference between gender (page 9, lines 375-381).

We appreciate you taking the time to offer us your insights related to the paper. We found your feedback very constructive. We tried to be responsive to your concerns.